# Determination of β-Agonists in Urine Samples at Low µg/kg Levels by Means of Pulsed Amperometric Detection at a Glassy Carbon Electrode Coupled with RP-LC

Annalisa Mentana [1,2], Carmen Palermo [1,3,*] and Diego Centonze [1,4]

1 Dipartimento di Scienze Agrarie, Degli Alimenti e Dell'ambiente, Università Degli Studi di Foggia, Via Napoli 25, 71122 Foggia, Italy; annalisa.mentana@unifg.it (A.M.); diego.centonze@unifg.it (D.C.)
2 Istituto Zooprofilattico Sperimentale Della Puglia e Della Basilicata, Via Manfredonia 20, 71122 Foggia, Italy
3 Dipartimento di Medicina Clinica e Sperimentale, Università Degli Studi di Foggia, Via Napoli 25, 71122 Foggia, Italy
4 Dipartimento di Scienze Mediche e Chirurgiche, Università Degli Studi di Foggia, Via Napoli 25, 71122 Foggia, Italy
* Correspondence: carmen.palermo@unifg.it; Tel.: +39-0881-589239

**Featured Application: A new method based on RP-HPLC with pulsed amperometric detection (PAD) of beta-agonists, synthetic drugs, has been developed. This method provided an efficient detection of analytes with high response sensitivity and represents a valid alternative to the conventional ones as demonstrated by the accurate analysis of urine samples.**

**Abstract:** A method for the determination of β-agonists was developed by combining the separation of analytes through high-performance liquid chromatography, with a reversed-phase column, coupled to the pulsed amperometric detection at a glassy carbon electrode. Preliminary experiments, using cyclic voltammetry, allowed for an understanding of the electrochemical behavior of clenbuterol, fenoterol, and terbutaline. By analyzing the electrochemical response, the conditions for detecting the analytes and for cleaning the working electrode were identified. The proposed potential-time profile was designed to prevent contamination of the carbon electrode following consecutive analyses, so ensuring a reproducible and sensitive quantitative determination. The waveform electrochemical parameters, including detection and delay times, have been optimized in terms of sensitivity, detection limits, and long-term response stability. The chromatographic separation was carried out using a C8 column in isocratic mode, and a mixture of acetic acid and acetonitrile. The optimized experimental conditions were used for the analysis of standard solutions and real samples. Detection limits, lower than the maximum residue limit set for clenbuterol by European directives, were obtained for all β-agonists investigated. The method validation was performed by evaluating the linearity, selectivity, precision, and recovery. Calf urine samples were used to verify the applicability of the proposed method, analyzing both enriched and naturally contaminated urine samples.

**Keywords:** beta-agonists; clenbuterol; reversed phase chromatography; pulsed amperometric detection; glassy carbon; calf urine

## 1. Introduction

Beta-adrenoceptor agonists (β-As) are derivatives of phenylethanolamine with different substituents on the aromatic ring and terminal amino group.

These compounds have been commonly used as a bronchodilator medicine for the treatment of human chronic obstructive pulmonary disease and asthma [1–3].

However, some of these compounds may be misused as performance-enhancing drugs in athletes to enhance performance due to their ergogenic potential and anabolic effects [4].

Another prohibited application of these substances is their use as veterinary drugs; in fact, they have been widely used in livestock to improve the transformation of the level of fat into the production of muscle proteins [5,6]. These illegal uses have harmful effects on human health since beta-agonist residues can accumulate in animal tissues and organs and may have a pharmacological effect in human, such as dizziness, headache, hypokalemia, nausea, tachycardia, cardiac QT interval changes, and effects in the central nervous system such as nervousness and difficulty sleeping [7].

The administration of ß-agonists as growth-promoting agents in food-producing animals is banned in many countries [8]. However, ractopamine and zilpaterol are authorized for production of some animals in a limited number of countries (e.g., the USA, Canada, and Brazil). Maximum residue limits (MRL$_S$) for ractopamine in cattle and pig muscle (both 10 µg/kg) have been adopted by the Codex Alimentarius Commission and have been implemented in some countries (e.g., Canada). Others have set alternative limits for ractopamine (e.g., USA-30 and 50 µg/kg in cattle and pig muscle). However, many countries have banned meat products containing ß-agonists. In the EU, ß-agonists other than clenbuterol (for cattle and horse) are prohibited for use in food producing animals. Therefore, the European Union has improved the official check programs to hinder the use of beta-agonists in stock farming and a maximum residue limit (MRL) of 0.5 µg/kg has been established for clenbuterol [9].

In the literature there are various analytical methods for the determination of beta agonists, among these are immunoenzymatic methods [10,11], gas chromatography [12,13], and HPLC separations, coupled with mass spectrometry [14,15], UV detection [16,17], or electrochemical detection [18,19]. However, these procedures are often lengthy, requiring large volumes of organic solvents and derivatization steps prior to gas chromatography analysis or laborious sample preparation prior to LC-MS determinations [20–22], in order to minimize interferences and the matrix effect on ion suppression. Electrochemical detection methods (ECDs), and in particular the pulsed amperometric detection, PAD-based ones [23–25], have attracted much interest as detection methods in HPLC. In this context, the main point to be considered in the development of such methods is the perfect tune between the mobile phase, also acting as the supporting electrolyte, and the electrode material/electrochemical technique used for the detection of analytes. As a consequence, organic mobile phases used in RP-HPLC are not really compatible with ECD at metal-based electrodes.

Hydroxy- or amino-phenyl moiety of beta agonists (see Figure 1) implies their electrochemical detection by anodic oxidation, and among the most common electrodes, the carbonaceous ones (including glassy carbon, GCE) were used [26–29].

With respect to metal electrodes, GCEs possess excellent electrochemical properties in a wide range of working potentials in organic solvents, but they undergo fouling phenomena when operating at constant potential with certain analytes, with a time-dependent deterioration of the electrochemical response. To overcome this limitation, we propose for the first time the proof-of-concept of PAD at GCEs [30]. The proposed approach allowed the development of a RP-HPLC/PAD method for polyphenols determination in real samples [31].

This work describes a novel approach for the detection of certain beta agonists, such as clenbuterol, terbutaline, and fenoterol. The method is based on the electrochemical oxidation of the analytes by applying a three-potential waveform at a glassy carbon working electrode. Taking into account that each analyte could have different electrochemical properties at GCE, depending on the electroactive functional groups and the organic mobile phase, a proper electrochemical characterization and optimization of the PAD waveform parameters have been carried out in order to obtain the best detection performances. Compounds detected in this way were previously separated by reverse phase high performance liquid chromatography and the chromatographic conditions have been carefully investigated.

The method can therefore be considered a good screening method since electrochemical detection is more selective than UV detection. A simple solid-phase extraction (SPE), a separation by means of a C8 column and the electrochemical detection ensure a qualitative and quantitative response suitable for the different analytes.

The method can consequently be applied to the rapid analysis of beta agonists in various samples of biological origin since it does not require the derivatization of the analytes or extensive sample pretreatment.

## 2. Materials and Methods

### 2.1. Chemicals and Working Standard Solutions

Clenbuterol, fenoterol, terbutaline, ultrapure water, methanol, and acetonitrile for HPLC were purchased from Sigma-Aldrich (Stenheim, Germany). Sodium hydroxide were purchased from J.T. Baker (Deventer, The Netherlands), while glacial acetic acid was from Carlo Erba Reagenti (Milano, Italy). Stock solutions (500 µg/L) of each β-agonist was prepared in methanol and stored at 4 °C in the dark. Working standard solutions, in the range 0.5–100 µg/L, were prepared by dilution with mobile phase just before use.

### 2.2. Sample Preparation

The sample clean-up of calf urine was performed by solid phase extraction (SPE) using a reverse-phase C18 tube (Supelclean™ LC-18 SPE, Supelco). Calf urine samples were centrifuged at 3000 rpm and filtered, and then an aliquot of 4 mL was applied to a C18 column, preconditioned by a sequential treatment with 5 mL of MeOH and 3 mL of water. After washing with 6 mL of water, the fraction containing the β-agonists was eluted with 3 mL of MeOH. The eluate was evaporated to dryness at 50 °C, under a nitrogen stream. Finally, the residue was solubilized in 400 µL of mobile phase (concentration factor of 10), and then injected.

### 2.3. Instruments and Method

Chromatographic separations were performed on a LC20 Chromatography Enclosure System (Dionex Corporation, Sunnyvale, CA, USA) composed of a GP50 gradient pump, a Rheodyne injection valve with a 25-µL injection loop, and an ED50 electrochemical detector. The flowthrough amperometric detection cell is made from a 1.0-mm diameter glassy carbon working electrode (GCE) and a standard combination pH-Ag | AgCl reference electrode, set to Ag mode; the titanium body of the cell served as the counter electrode. Separations were performed using a C8 column LiChrospher$^{®}$ 60 RP-select B (25 cm × 4 mm, 5 µm I.D. (Merck KGaA, Darmstadt Germany) coupled with the guard column LiChroCART LiChrospher 60 RP-select B (4 mm × 4 mm, 5 µm I.D., Merck) at a flowrate of 1.0 mL/min by isocratic elution with 20 mM of acetic acid/acetonitrile (55:45 $v/v$). The pH was adjusted to 7.0 with acetic glacial acid or sodium hydroxide, as necessary. The reservoir bottles were closed and pressurized with pure nitrogen to 0.8 MPa. A three-step waveform PAD was employed which consists of three potentials and four-time parameters. The optimized applied potentials and pulse durations were the following: $E_{OX}$ = +2.0 V ($t_{OX}$ = 50 ms), $E_{DET}$ = +1.2 V ($t_{DEL}$ = 100 ms, and $t_{INT}$ = 800 ms), and $E_{RED}$ = −2.0 V, ($t_{RED}$ = 50 ms). The system was interfaced, via proprietary network chromatographic software (PeakNet™, Dionex Corporation, Sunnyvale, CA, USA), to a personal computer, for instrumentation control, data acquisition, and processing.

Cyclic voltammetry (CV) studies were carried out by a potentiostat CHInstruments Electrochemical Analyzer and CHI620A software version 2.07 (Austin, TX, USA). Batch apparatus consisted of a conventional three electrode cell, equipped with a Pt wire counter electrode, an Ag/AgCl (saturated KCl) reference electrode, and a glassy carbon (3 mm diameter) as the working electrode. Prior to use, the surface of the GCE was polished with emery paper and alumina powder, then, ultrasonicated. The supporting electrolyte (20 mM acetic acid/acetonitrile (55:45 $v/v$) at pH 7) was deaerated by nitrogen bubbling for 10 min.

### 2.4. Validation Procedure

A linearity test was performed by the regression line of data obtained throughout three series of analyses on three different days, by injecting standard solutions of beta-agonists, in the range 0.5–100 µg/L. The method selectivity was tested by the analysis of 20 independent blank samples of calf urine, found negative by Enzyme-linked immune assay screening (ELISA). The absence of interfering peaks in the retention time-window of interest was checked for each analyte within the ±2.5% retention time range. Precision and recovery data were obtained from the analysis of calf urine samples fortified, prior to the extraction, with clenbuterol in the concentration range 0.5–20 µg/kg. The experiments were performed in different days with the same instruments but different operators and instrumental calibrations.

## 3. Results and Discussion

### 3.1. Voltammetric Studies

By cyclic voltammetry (CV) [32–34], the electrochemical responses of β-agonists (see Figure 1) at a glassy carbon electrode (GCE) were studied in 20 mM of acetic acid/acetonitrile (55:45 *v/v*) at a pH 7 solution as the supporting electrolyte, to mimic a typical chromatographic mobile phase. The current–potential (i–E) curves of clenbuterol are shown in Figure 2A,B. To explore the effect of the electrochemical process, a series of voltammograms corresponding to two different starting potentials (−1.5 and 0.0 V) are presented. From the comparison of clenbuterol CVs (blue and red solid lines) compared to blank (black line), some important features can be noticed. The electrochemical behavior of clenbuterol in the explored range of potentials, from the negative (−1.5 V) to the positive limit (+1.5 V), shows a higher peak current (+0.92 V, oxidation of the aniline group) than that obtained in a positive range of potential (0.0 V–1.5 V). This phenomenon suggests the involvement of the reduced electrode surface in the oxidation mechanism. Moreover, a strong current response decrease can be observed, when a positive interval of potentials is used; for instance, in the second cycle (data not shown), the current is almost half of that of the first cycle and it becomes less than 40% in the tenth. This phenomenon is minimal when the experiment is carried out in the extended potential range (−1.5 V/+1.5 V). A similar voltammetric response was observed for the other β-agonists; see for instance the CVs of terbutaline in Figure 3A,B. The oxidation, attributed to the terbutaline phenolic groups, occurs at a very close potential (1.03 V) of that observed for clenbuterol, showing that GCE in acetic/acetonitrile medium is suitable for the simultaneous electrochemical detection of β-agonists. In addition, CV experiments evidenced that β-agonists have the advantage/drawback of adsorbing on glassy carbon electrodes. The adsorption process is occasionally desirable since it provides a chance to increase the analytical response by pre-concentrating, but it also means that fouling of the working electrode occurs by the aromatic analytes as well as their oxidation products. These results suggest that a cleaning before each subsequent electrochemical measurement is necessary, therefore, a well-designed triple potential step waveform would be suitable to detect β-agonists following liquid chromatographic separations with organic mobile phases.

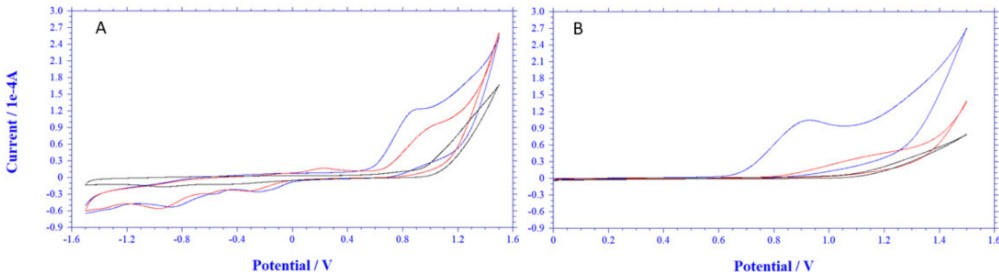

**Figure 1.** Chemical structures of clenbuterol, fenoterol, and terbutaline.

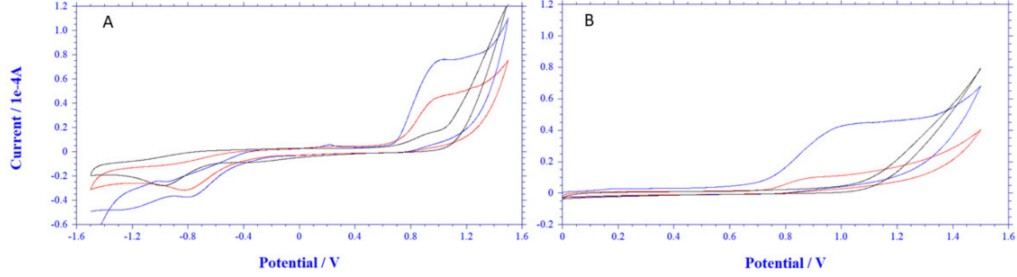

**Figure 2.** Current-potential curves of 5 mM of clenbuterol (blue line 1st cycle, red line 10th cycle) at a glassy carbon electrode in 20 mM of acetic acid/acetonitrile (55:45 *v/v*) at pH 7.0 supporting electrolyte (black line). Scan rate: 100 mV.s$^{-1}$. Potential range: (**A**) −1.5/+1.5 V; (**B**) 0/+1.5 V.

**Figure 3.** Current-potential curves of 5 mM of terbutaline (blue line 1st cycle, red line 10th cycle) at a glassy carbon electrode in 20 mM of acetic acid/acetonitrile (55:54 *v/v*), pH 7.0 supporting electrolyte (black line). Scan rate: 100 mV.s$^{-1}$. Potential range: (**A**) −1.5/+1.5 V; (**B**) 0/+1.5 V.

### 3.2. Triple Potential Step PAD Waveform Optimization

Within a pulsed waveform, duration, and potential values of each step may significantly affect the sensitivity and reproducibility of response. Thus, to further increase the detection capability of the triple step PAD waveform, a systematic investigation was carried out to optimize the potential-time settings by varying each value, while all the other parameters were held constant. Waveform optimization was carried out by performing multiple injections of the same mixed standard solution of β-agonists in a C8 column,

under isocratic elution conditions at 1.0 mL/min. The potential waveform was applied to the glassy carbon working electrode, and the choice of the optimal potential-time values providing the maximum sensitivity was performed by the evaluation of the signal-to-noise ratio (average of three replicates) as a function of the parameter to be optimized. For each set of applied potential, the background noise was evaluated as the peak-to-peak value. The detection potential was increased from +0.8 to +1.3 V in 0.05 V intervals and the peak current increased as the potential increased, reaching a maximum at 1.2 V. Subsequent increments produced a higher noise, so +1.2 V was selected as the optimum detection potential. The effect of integration times was studied in the range 150–800 ms. Higher values are to be avoided to keep the overall cycle time short, enough to maintain peak integrity. It was observed that integration times lower than 800 ms determined a sensitivity decrease. Potential values of +2.0 V and −2.0 V, respectively, for the steps of oxidative cleaning and reductive electrode restoration, were essential for good response stability, ensuring the oxidation of the surface (to avoid the electrode fouling) and the reactivation of working electrode. The corresponding time parameters were varied from 10 to 200 ms. A value of 50 ms both for $t_{OX}$ and $t_{RED}$ gave the best results, since higher times produced a substantial decrease in the response. The potential-time parameters selected as the optimum are reported in Table 1. In order to demonstrate the efficiency of PAD to preserve the signal magnitude, repetitive injections of spiked calf urine sample were performed. No significant difference was found between the initial and final readings (*t*-test, 95% confidence, *n* = 12). The optimized waveform was also evaluated in terms of detection limit, using clenbuterol as the model compound. A LOD of 0.1 μg/kg was obtained, with respect to 0.4 μg/kg resulted by DC analysis. As far as we know, the LOD of the present method is approximately 40–100 fold lower than the values obtained by HPLC-UV [6,35] and comparable (even slightly lower) with the detection capability of 0.15 ppb determined for clenbuterol in bovine muscle by the LC-MS screening method [36].

**Table 1.** Optimized potential-time parameters of the triple-step PAD waveform.

| | Potential (V vs. Ag\|AgCl) | | | Time (ms) | |
|---|---|---|---|---|---|
| Parameter | Explored Value | Optimized Value | Parameter | Explored Value | Optimized Value |
| $E_{DET}$ | +0.8 to +1.3 | +1.2 | $t_{DET}$ | 200 to 900 | 900 |
| | | | $t_{DEL}$ | 50 to 100 | 100 |
| | | | $t_{INT}$ | 150 to 800 | 800 |
| $E_{OX}$ | +2.0 | | $t_{OX}$ | 10 to 200 | 50 |
| $E_{RED}$ | −2.0 | | $t_{RED}$ | 10 to 200 | 50 |

### 3.3. Method Validation and Application to Real Samples

The method was submitted to a preliminary validation process and the analytical performances of linearity, selectivity, precision, and recovery have been determined by the analysis of blank and spiked calf urine samples.

In order to verify the absence of matrix interfering peaks in the retention time-window of interest (±2.5% of the retention time of each β-agonist), 20 independent calf urine samples were analyzed, and an example of chromatogram of a blank urine sample is shown in Figure 4 and compared with that of a standard mix of clenbuterol, terbutaline, and fenoterol. The selectivity of the method toward endogenous compounds is demonstrated by the absence of interfering peaks at the retention times of the β-agonists.

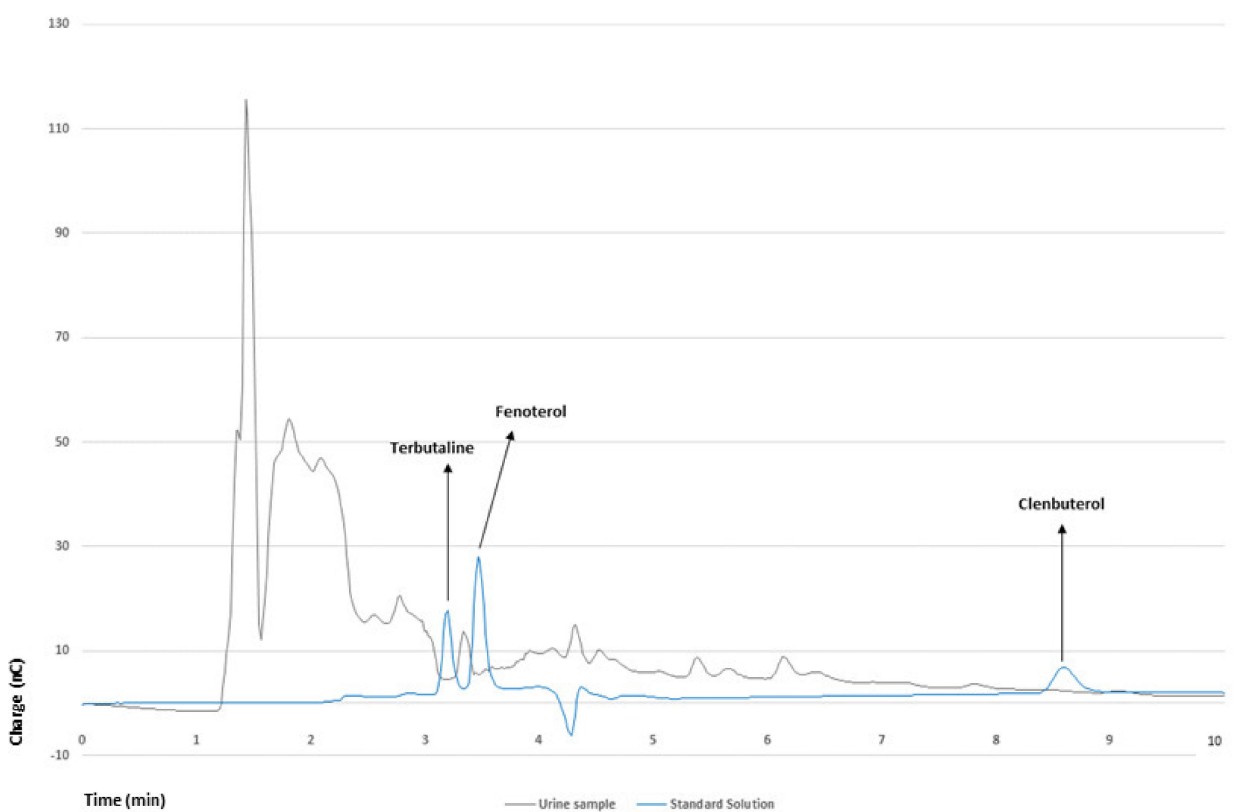

**Figure 4.** Comparison of chromatograms of a mixed standard solution of β-agonists, each at a concentration of 100 μg.L$^{-1}$ (blue) and a calf urine sample purified by SPE (grey). Column LiChroCART C8 LiChrospher 60 RP-select B column (25 cm × 4 mm, 5 μm I.D., Merck) coupled with a guard column LiChroCART LiChrospher 60 RP-select B (4 mm × 4 mm, 5 μm I.D., Merck). Mobile phase: 20 mM of acetic acid/acetonitrile (55:45), pH 7.0. Flow rate 1.0 mL.min$^{-1}$. Injection volume: 25 μL. Detection: PAD at a glassy carbon electrode.

Table 2 shows reported data relevant the performance parameters obtained for all the investigated analytes. A good linearity was found for the β-agonists determination in the range 0.5–100 μg/L, with correlation coefficients higher than 0.9984. The goodness of fit of the data to the calibration curve is obtained in terms of response factor distribution (signal to concentration ratio, yi/xi) whose reference range is (y/x) mean ±10%. Furthermore, any systematic instrumental bias can be ruled out since the intercept includes the zero value at a 95% confidence level. From the regression line parameters of the calibration curves obtained from standard solutions and spiked blank urine samples, instrumental and method LODs and LOQs have been calculated as 3.3 $s_{y/x}$/S and 10 $s_{y/x}$/S respectively, where S is the slope and $s_{y/x}$ is the standard deviation of residuals. The values were in the range 0.1–0.2 μg/L and 0.2–0.5 μg/L, respectively, which allow a reliable quantification considering the MRL of 0.5 μg/kg, established for clenbuterol.

The method was tested for intraday precision by replicate injections of a spiked calf urine sample at 100 μg/L, and the performance parameters in terms of signal-to-noise ratio (S/N) and relative standard deviations (RSD) of 10 replicate injections of the analytes are given in Table 3. The RSD down to 7% demonstrate that PAD assures good signal reproducibility. For comparison, Table 3 also shows data obtained under DC electrochemical detection at +1.2 V, pointing out the response decrease that occurs when repeated injections are performed at a constant potential (RSD higher than 13%). The lower response stability for constant potential detection is also confirmed by a higher variability of the noise levels and concomitant lower signal-to-noise ratios with respect to PAD. Recoveries of β-agonists were evaluated by comparing the concentration of spiked samples determined by interpolation on the calibration curve with the nominal concentration of the fortification level. As an example, Figure 5 reports the comparison of a spiked calf

urine sample at 10 µg/L of each analyte and a standard solution at 100 µg/L. Considering the 10-fold concentration factor introduced by the SPE purification step, a very good recovery is displayed, as well as a satisfactory separation of terbutaline and fenoterol from an interference peak, which was present in all blank urine samples. Table 4 displays the recovery values obtained for all the analytes investigated at three spiking levels in the range 0.5–20 µg/L. The results in the range 78.6–97.2% are in agreement with the reference values indicated by the European Commission for the validation of analytical methods [37].

**Table 2.** Performance parameters of β-Agonists analyzed by LC-PAD at GCE.

| β-Agonist | Sensitivity | iLOD [a] | iLOQ [a] | Linear Range | R | mLOD [b] | mLOQ [b] |
|---|---|---|---|---|---|---|---|
| | µC/ppb | | | µg/L | | | µg/L |
| TERBUTALINE | $3.2 \times 10^{-4}$ | 0.23 | 0.50 | 0.5–100 | 0.9984 | 0.31 | 0.64 |
| FENOTEROL | $1.7 \times 10^{-4}$ | 0.14 | 0.22 | 0.5–100 | 0.9984 | 0.19 | 0.38 |
| CLENBUTEROL | $4.9 \times 10^{-5}$ | 0.10 | 0.20 | 0.5–100 | 0.9993 | 0.12 | 0.36 |

[a] Instrumental limits obtained by the regression parameters of calibration curves obtained from the analysis of standard solutions in the concentration range 0.5–20 µg/L. [b] Method limits obtained by the regression parameters of calibration curves obtained from the analysis of spiked blank samples in the concentration range 0.5–20 µg/L.

**Table 3.** Signal to noise ratio (S/N) and relative standard deviation of β-Agonists evaluated for PAD and DC detections in LC at GCE.

| Parameter | β-Agonist | Detection Method | |
|---|---|---|---|
| | | PAD | DC |
| S/N [a] | TERBUTALINE | 340 ± 200 | 340 ± 250 |
| | FENOTEROL | 310 ± 200 | 280 ± 200 |
| | CLENBUTEROL | 62 ± 37 | 95 ± 63 |
| RSD [b] | TERBUTALINE | 7.0% | 13.0% |
| | FENOTEROL | 3.6% | 14.0% |
| | CLENBUTEROL | 4.0% | 17.9% |

[a] Average signal to noise ratio (S/N), $n = 10$. [b] Relative standard deviation (RSD) for 10 successive injections of a standard solution of β-agonists at a concentration of 100 µg/L each.

**Table 4.** Recoveries from urine calf samples spiked with β-Agonists at three different levels.

| β-Agonist | Spiked Level | Mean Recovery ± SD [a] |
|---|---|---|
| | µg.L$^{-1}$ | % |
| TERBUTALINE | 0.5 | 79.1 ± 9.5 |
| | 5.0 | 81.5 ± 8.6 |
| | 20.0 | 95.3 ± 9.5 |
| FENOTEROL | 0.5 | 83.5 ± 10.0 |
| | 5.0 | 90.5 ± 9.1 |
| | 20.0 | 97.2 ± 8.9 |
| CLENBUTEROL | 0.5 | 78.6 ± 8.6 |
| | 5.0 | 92.1 ± 9.2 |
| | 20.0 | 93.4 ± 8.4 |

[a] Standard deviation (SD), $n = 3$.

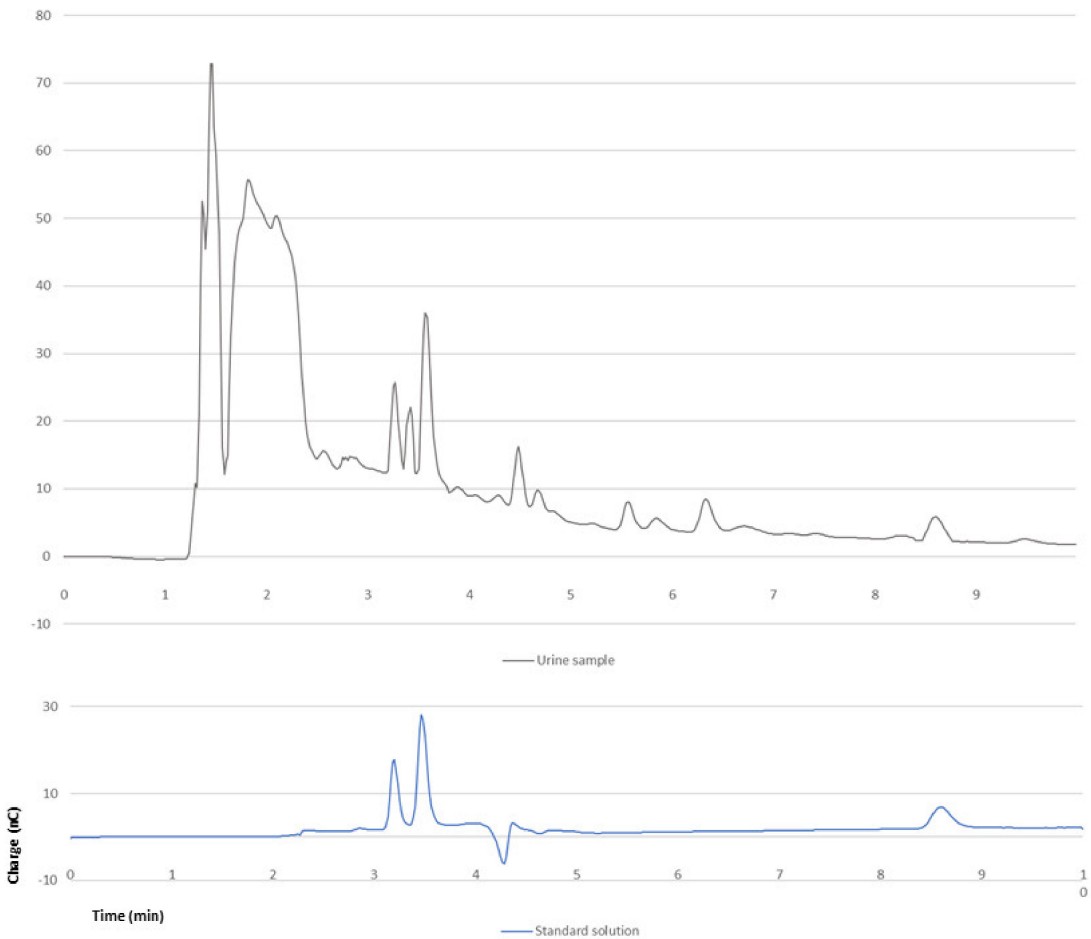

**Figure 5.** Chromatographic separations of a mixed standard solution of β-agonists, each at a concentration of 100 μg.L$^{-1}$ (blue) and a calf urine sample spiked at 10 μg.L$^{-1}$ (grey) and purified by SPE (10 fold concentration factor). Experimental conditions as in Figure 4.

The applicability and usefulness of the proposed method has been demonstrated by the analysis of blank and contaminated calf urine samples. Quantification was performed with the standard addition method. The quantitative determinations have been compared with the results obtained by the ELISA method (see Table 5), confirming the feasibility of the RP-LC/PAD method for quantitative screening purposes.

**Table 5.** Clenbuterol determination in calf urine samples by RP-LC-PAD and ELISA.

| Calf Urine Sample | Clenbuterol Content (μg/kg) | |
|---|---|---|
| | **RP-HPLC/PAD** | **ELISA** |
| 1 | ND [a] | negative |
| 2 | 0.45 [b] | negative |
| 3 | 1.5 | positive |

[a] Not Detectable, content < LOD. [b] Content lower than MRL (0.5 μg/kg).

## 4. Conclusions

A new method based on reverse-phase liquid chromatography and pulsed ampero-metric detection at a glassy carbon electrode has been developed for the determination of clenbuterol, fenoterol, and terbutaline in calf urine samples. A triple waveform is suggested that greatly improves the response sensitivity and stability over previous electrochemical detection techniques. The proposed method provided an efficient detection of beta-agonists with good long-term reproducibility that confirms the elimination of electrode fouling, occurring when electrochemical detection at a constant potential is employed. Wide linear

ranges of response and very good chromatographic parameters have been obtained for all the investigated beta-agonists. Detection limits resulted in the range 0.1–0.2 µg/kg and injection-to-injection repeatability (*n* = 10) was better than 7%. The obtained analytical results demonstrate that PAD at GCEs following chromatographic separations could be successfully applied for simultaneous determination of β-agonists in complex biological samples, at quantification levels of 0.2–0.5 µg/kg that resulted better than previous ECD or UV methods coupled with separation [6,18,19,26,27,35] or almost comparable to those of MS-based methods [11,20–22,36]. Moreover, the obtained recoveries were very good, well-complying with requirements indicated by the European Commission for the validation of analytical methods [37]. Therefore, the proposed RP-LC/PAD method can be considered a valid alternative to the conventional LC-UV and ELISA methods for a quantitative screening in routine analyses.

**Author Contributions:** A.M.: Investigation. C.P.: Investigation, validation, formal analysis, visualization, writing review & editing. D.C.: Conceptualization, supervision, methodology, writing review & editing. All authors have read and agreed to the published version of the manuscript.

**Funding:** This research received no external funding.

**Institutional Review Board Statement:** Not applicable.

**Informed Consent Statement:** Not applicable.

**Conflicts of Interest:** The authors declare no conflict of interest.

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
