# Peer review of "Determination of β-Agonists in Urine Samples at Low µg/kg Levels by Means of Pulsed Amperometric Detection at a Glassy Carbon Electrode Coupled with RP-LC"

_applsci, doi:10.3390/app112311302_

Round 1

Reviewer 1 Report

The authors present a novel method for detection of beta-agonists in bovine urine samples. The novelty lies in utilzing an electrochemical pulsed amperometric detection method for quantitative determination of beta-agonsit cocnentration in biological samples as opposed to UV or ELISA detection. Prior to analysis, the pristine biological samples were purified using revers phase HPLC.

All in all, the authors clearly explain the benefits of the method as compared to other techniques and present very sound data verifying their initial assumptions. However, what is left missing in the introduction and in the discussion of the result is the actual comparison of other electrochemical detection techniques or previous work using PAD for otehr analytes. As such, the use of PAD in HPLC is not new neither is the use of electrochemical detection for beta-agonist quanitifaction. So, the authros should better present the state of the art, not only in non-electrocemical techniques but also in electrochemical ones, and how their novel method surpasses the current performance standards of these approaches.

The authors should go over the text to clean up the text left over from the paper template.

Author Response

Response to Reviewer 1

The authors present a novel method for detection of beta-agonists in bovine urine samples. The novelty lies in utilzing an electrochemical pulsed amperometric detection method for quantitative determination of beta-agonsit cocnentration in biological samples as opposed to UV or ELISA detection. Prior to analysis, the pristine biological samples were purified using revers phase HPLC.

All in all, the authors clearly explain the benefits of the method as compared to other techniques and present very sound data verifying their initial assumptions. However, what is left missing in the introduction and in the discussion of the result is the actual comparison of other electrochemical detection techniques or previous work using PAD for otehr analytes. As such, the use of PAD in HPLC is not new neither is the use of electrochemical detection for beta-agonist quanitifaction. So, the authros should better present the state of the art, not only in non-electrocemical techniques but also in electrochemical ones, and how their novel method surpasses the current performance standards of these approaches.

Answer: ECD based methods, and in particular the PAD based ones, have been extensively exploited as detection methods in HPLC. In this context, the main point to be considered in the development of such methods is the perfect tune between the mobile phase, also acting as the supporting electrolyte, and the electrode material/electrochemical technique used for the detection of analytes. As a consequence, organic mobile phases used in RP-HPLC are not really compatible with ECD at metal based electrodes. On the contrary, carbonaceous electrodes (including GCEs) possess excellent electrochemical properties in a wide range of working potentials in organic solvents, but these electrodes undergo fouling phenomena, when operate at constant potential, with a time-dependent deterioration of the electrochemical response. To overcome this limitation, we propose for the first time the proof-of–concept of PAD at GCEs (see Analytica Chimica Acta, 894 (2015) 1-6, now ref. 30). The proposed approach allowed the development of a RP-HPLC/PAD method for polyphenols determination in real samples (see Journal of Chromatography A, 1420 (2015) 66–73, now ref. 31). Taking into account that each analyte could have different electrochemical properties at GCE, depending on both the electroactive functional groups and the supporting electrolyte (organic mobile phase), a proper electrochemical characterization and optimization of the PAD waveform parameters are required in order to obtain the best detection performance. Therefore, as suggested by the Reviewer, the introduction and conclusions have been modified as follow, to better evidence the improvements of the proposed method with respect to the state of the art.

The authors should go over the text to clean up the text left over from the paper template.

Answer: Text modifications have been carried out accordingly.

Reviewer 2 Report

This paper deals with the method for determination of β-agonists in urine samples and given the signifacance of analites in veterinary medicine can have a high practical value. 

On the other hand, the paper has ambiguities and shortcomings that I will list:

- As can be seen in Figure 4, the urine sample is not clean enough at the time of the elution of terbutaline and fenoterol. So it seems that the selection of the sample preparation method is not the best. It is not explained nor is it optimized.

- In Figure 5, the sample was spiked only with clenbuterol? If yes, please provide the chromatogram of sample spiked with all tested β-agonists.

-Please provide a table with recovery values specifically for each compound at every tested concentration

-Please explain the selection of C8 column and mobile phase.  Why not C18 or normal phase column?

Author Response

Response to Reviewer 2

This paper deals with the method for determination of β-agonists in urine samples and given the signifacance of analites in veterinary medicine can have a high practical value.

On the other hand, the paper has ambiguities and shortcomings that I will list:

- As can be seen in Figure 4, the urine sample is not clean enough at the time of the elution of terbutaline and fenoterol. So it seems that the selection of the sample preparation method is not the best. It is not explained nor is it optimized.

Answer: Urine is one of the most complex matrices being plenty of interferences, and in preliminary experiments (data not shown) we tested 4 different conventional sample clean-up methods: SPE-C18, SupelMIPTM SPE, IAC, and Bond Elut Plexa PCX. SPE-C18 and SupelMIPTM SPE gave clean extract in the elution time window of clenbuterol, but only SPE-C18 showed good results for terbutaline and fenoterol, even if an interference was present between the two analyte. In all the other cases, several interferents were present in the time window of the analyte making impossible an accurate quantification.

- In Figure 5, the sample was spiked only with clenbuterol? If yes, please provide the chromatogram of sample spiked with all tested β-agonists.

Answer: The required chromatogram now has been displayed in figure 5.

-Please provide a table with recovery values specifically for each compound at every tested concentration

Answer: A table (now table 4) with recoveries at the tested concentration for each analyte has been added.

-Please explain the selection of C8 column and mobile phase.  Why not C18 or normal phase column?

Answer: Urine is one of the most complex matrices being plenty of interferences, and the use of a C18 column (in preliminary experiments we tested a Discovery Supelco 5 μm, 250 mm x 4.6 mm with an without SDS as ion pair reagent) did not allow the separation of interferences from the analytes of interest, in particular, in the range of elution times of fenoterol and terbutaline.

Reviewer 3 Report

Reprepare Figures 2-5. Reduce numbers indicated on horizontal, and use large size fonts

mg .L-1, mg.kg-1, 1.0 mL.min-1

Author Response

Response to Reviewer 3

Reprepare Figures 2-5. Reduce numbers indicated on horizontal, and use large size fonts

mg .L-1, mg.kg-1, 1.0 mL.min-1

Answer: Modifications have been carried out accordingly.

Reviewer 4 Report

The paper deals with a new method for the determination of three beta-agonists in urine using liquid chromatography with electrochemical detection. Although the work may be of interest to the readers of Applied Sciences some points should be addressed.

General comments

The performance of the proposed method should be compared with others reported in the scientific literature, especially with those based on LC-MS, which is probably the technique of choice. From a preliminary search, I have found several references displaying pretty good analytical figures of merit. Besides, some interesting reviews on the topic should be cited and commented on.

The number of selected analytes is limited and several relevant drugs such as salbutamol have not been considered. Why?

To summarize, the authors should clarify the pros and cons of this method. A critical discussion of this issue is essential since, otherwise, the novelty of the paper and its contribution to the field will remain unclear.

Specific comments

Please revise hyphens for correctness (e.g., dizziness, continuous, produce, and many others).

Experimental section: concentrations of the stock solutions should be indicated. The concentration range of working solutions should be given as well.

2.3 Section title: Instruments should be written instead of Apparatus

Line 165. The support electrolyte composition should be given (I guess it is the same indicated in line 145).

Figures 1 and 4 have not been introduced in the text.

Figure 4 should be improved. For instance, X and Y axis definitions and units are missing (e.g. Time (min)); the format of numbers is wrong (do use dots); frames around the graphic should be removed; X-axis step could be 1 or 2 min.

Figure 5. Use the same format as in Fig. 4. Y-axis is missing.

The beta symbol is wrong in several places (e.g. Fig. 5, Table 2, and Table 3). Please check the whole manuscript.

Table 3. The meaning of data such as 340±199 is unclear and it should be better explained. Besides, if it refers to an experimental value and its standard deviation, it should be written as 300±200.

Line 242. Please explain why LOQs of 0.2 to 0.5 µg L-1 comply with the MRL of 0.5 µg Kg-1.

References should be given according to the journal format.

Author Response

Response to Reviewer 4

The paper deals with a new method for the determination of three beta-agonists in urine using liquid chromatography with electrochemical detection. Although the work may be of interest to the readers of Applied Sciences some points should be addressed.

General comments

The performance of the proposed method should be compared with others reported in the scientific literature, especially with those based on LC-MS, which is probably the technique of choice. From a preliminary search, I have found several references displaying pretty good analytical figures of merit. Besides, some interesting reviews on the topic should be cited and commented on.

Answer: The proposed screening quantitative method was already compared to others from the literature (see lines 299-302), but perhaps in a partial way. LC-MS is certainly the “technique of choice” for confirmation analysis, but extensive sample pre-treatments are required in order to minimize interferences and the matrix effect on the ion suppression. This phenomenon affects many aspects of the performance of the method, such as LOD and accuracy, as reported in recent reviews on this topic, which, as suggested by the Reviewer have been quoted (A.A.M. Stolker, U.A.Th. Brinkman, Analytical strategies for residue analysis of veterinary drugs and growth-promoting agents in food-producing animals—a review Journal of Chromatography A, 1067 (2005) 15–53; H. F. De Brabander, B. Le Bizec, G. Pinel, J.-P. Antignac, K. Verheyden, V. Mortier, D. Courtheyn and H. Noppe, Past, present and future of mass spectrometry in the analysis of residues of banned substances in meat-producing animals, Journal of Mass Spectrometry 2007; 42: 983–998;  A.G. Fragkaki, C. Georgakopoulos, S. Sterkc, M.W.F. Nielen, Sports doping: Emerging designer and therapeutic β2-agonists - Invited critical review, Clinica Chimica Acta 425 (2013) 242–258). Moreover, additional comparisons with other scientific papers from the literature and comments have been reported in the introduction and conclusions sections.

The number of selected analytes is limited and several relevant drugs such as salbutamol have not been considered. Why?

Answer: The main goal of the proposed method is the development of a sensitive and stable electrochemical detection method of analytes based on PAD at GCEs following their separation by RP-HPLC. Taking into account that each analyte could have different electrochemical properties at GCE, depending on both the electroactive functional groups and the supporting electrolyte (organic mobile phase), a proper electrochemical characterization and optimization of the PAD waveform parameters are required in order to obtain the best detection performance. Therefore, we selected target analytes belonging to the class of beta-agonists and possessing different electroactive functional groups: amino-phenyl (clenbuterol); hydroxy-phenyl (terbutaline) and two hydroxy-phenyl moyeties (fenoterol). The structure of salbutamol is very similar to that of terbutalin having a hydroxy-phenyl electroactive functional group.

To summarize, the authors should clarify the pros and cons of this method. A critical discussion of this issue is essential since, otherwise, the novelty of the paper and its contribution to the field will remain unclear.

Answer: As mentioned above, new references and discussions have been added in order to clarify the benefits and limitations of the proposed method with respect to the state of art.

Specific comments

Please revise hyphens for correctness (e.g., dizziness, continuous, produce, and many others).

Answer: Text modifications have been carried out accordingly.

Experimental section: concentrations of the stock solutions should be indicated. The concentration range of working solutions should be given as well.

Answer: Modifications have been carried out accordingly.

2.3 Section title: Instruments should be written instead of Apparatus

Answer: The suggested modification has been made.

Line 165. The support electrolyte composition should be given (I guess it is the same indicated in line 145).

Answer: The support electrolyte composition has been entered

Figures 1 and 4 have not been introduced in the text.

Answer: The figures have been correctly introduced in the text.

Figure 4 should be improved. For instance, X and Y axis definitions and units are missing (e.g. Time (min)); the format of numbers is wrong (do use dots); frames around the graphic should be removed; X-axis step could be 1 or 2 min.

Answer: Modifications have been carried out accordingly.

Figure 5. Use the same format as in Fig. 4. Y-axis is missing.

Answer: Modifications have been carried out accordingly.

The beta symbol is wrong in several places (e.g. Fig. 5, Table 2, and Table 3). Please check the whole manuscript.

Answer: Corrections were made accordingly.

Table 3. The meaning of data such as 340±199 is unclear and it should be better explained. Besides, if it refers to an experimental value and its standard deviation, it should be written as 300±200.

Answer: Data displayed in the first part of table 3 represent the mean signal-to-noise ratio and the relevant standard deviations for each analyte obtained from chromatograms of 10 replicated injections of a standard mix at 100 µg/L. These data give an idea about the variability of the noise level with respect to the variability of the signal showed in the lower part of the table 3 for the two different electrochemical techniques applied to a GCE, namely DC, the conventional one, and PAD. A higher variability of the noise levels in DC and concomitant lower signal-to-noise ratios with respect to PAD denote a lower response stability for constant potential detection. These data have been better explained as follow and the correct approximations have been carried out.

Line 242. Please explain why LOQs of 0.2 to 0.5 µg L-1 comply with the MRL of 0.5 µg Kg-1.

Answer: MRL is the maximum residue limit admitted in real samples by the law (see ref. 9), therefore a validated analytical method must have a LOQ below or at least equal to the law limit in order to allow a reliable quantification. The sentence has been rephrased to make clear this concept.

References should be given according to the journal format.

Answer: The references were revised according to the format of the journal.

Reviewer 5 Report

This paper describes the sensitive determination of three beta-agonists including clenbuterol, fenoterol and terbutaline in calf urine using high-performance liquid chromatography (HPLC) with pulsed amperometric detection.  The proposed method may be useful for the simultaneous determination of these three compounds present in biological samples with its sensitivity and selectivity.  However, the electrochemical detector (ECD) has been widely used for the detection of oxidative-reductive compounds in the HPLC analysis.  In the present study, the mechanical condition of the ECD detection is only optimized for three beta-agonists, which is naturally carried out for the HPLC analysis of the target compounds.  From this reason, the novelty of the present study is considered to be low so that the paper is not recommended to the publication in Applied Sciences.  The paper is also roughly completed with a lot of word hyphen mistakes and the collection of Figures and Tables at the end of the text (3.4 section), which should be improved to distribute to the suitable position between the text sentences. Detailed comments are described below:

  1. In the Introduction, it describes that the present method “does not require the derivatization of the analytes or extensive sample pretreatment (page 3, line 3 – 5). However, only three calf urine samples for clenbuterol were measured using the present method as summarized in Table 4, and these obtained values were not evaluated in the results and discussion section. Urine is easy to collect at large volumes so that the reason why the sensitive detection of these three compounds is required should be clearly explained except for the technical points.
  2. In the Results and discussion section, it describes that “20 independent calf urine samples have been analyzed” (page 5, line 230 – 231). However, the obtained data are not found in this paper.
  3. The recovery test is only applied to clenbuterol at the values of 78.6 – 93.4%, while no data on terbutaline and fenoterol are found in the paper. The experimental result of the recovery test for three target compounds should be clearly summarized in one table, because the simple pretreatment is proposed to the present method.
  4. In Figures 4 and 5, large amounts of hydrophilic compounds in the calf urine sample are eluted before or near terbutaline and fenoterol peaks, while the pretreatment using the C18 cartridge is completed. The impurity peak interrupts the terbutaline and the fenoterol peaks. This result requires further pretreatment procedures.
  5. Table 2 summarizes the validation data including LOD, LOQ, linear range and R value. In the HPLC chromatogram shown in Figures 4 and 5, different peak areas of three target compounds were found in each individual retention time. Increased peak area is detected in the order of fenoterol, terbutaline and clenbuterol, while same concentration of 100 µg/L for each compound is injected into the present HPLC system.  However, in Table 2 the LOD value of terbutetaline is lower than that of fenoterol and clenbuterol, where the LOD value of fenoterol is equal to that of clenbuterol.  Higher LOQ value is also listed in the order of fenoterol, clenbuterol and terbutaline, while each linear range is all same in these compounds.  These disagreements are not understandable for readers.  The range of significant figures should be considered to each value.
  6. Table 4 summarizes the comparison of analytical values of clenbuterol in calf urine between RP-HPLC/PAD and ELISA. However, the data obtained from ELISA is expressed by “negative” and “positive”, which may difficult to compare the analytical values between these two measurement systems. In the calf urine sample #2, the “negative” in ELISA is corresponded to 0.45 in RP-HPLC/PAD, and in the calf urine sample #3, the “positive” in ELISA is corresponded to 1.5 in RP-HPLC/PAD.  What is the value obtained by RP-HPLC/PAD corresponded to the lowest “positive” value obtained by ELISA?  As to terbutaline and fenoterol, similar experiments should be required.

Other comments

・page 1, line 33: Beta-agonists ---> beta-agonists

・page 1, line 34: Calf urine ---> calf urine

・page 2, line 47: dissi-ness ---> dissiness

・page 2, line 63: gas-chromatography ---> gas chromatography

・page 2, line 67: deter-minations ---> determinations

・page 2, line 68: sim-plicity ---> simplicity

・page 2, line 70: aryletanlaminic ---> arylethanolaminic

・page 2, line 70: fig. 1 ---> Figure 1

・page 2, line 71: am-pero- ---> ampero-

・page 2, line 77: detec-tion ---> detection

・page 2, line 78: prod-ucts ---> products

・page 2, line 83: sur-face ---> surface

・page 2, line 84: continu-ously ---> continuously

・page 2, line 85: amperom-etry ---> amperometry

・page 2, line 86: pro-duce ---> produce

・page 3, line 97: quali-ta- ---> qualita-

・page 3, line 106: acetic acid glacial ---> glacial acetic acid

・page 3, line 132: acetic glacial acid ---> glacial acetic acid

・page 3, line 142: witha ---> with a

・page 3, line 147: min.. (double periods) ---> min.

・page 8, Table 2: β-agonist ---> β-Agonist

Author Response

Response to Reviewer 5

This paper describes the sensitive determination of three beta-agonists including clenbuterol, fenoterol and terbutaline in calf urine using high-performance liquid chromatography (HPLC) with pulsed amperometric detection.  The proposed method may be useful for the simultaneous determination of these three compounds present in biological samples with its sensitivity and selectivity.  However, the electrochemical detector (ECD) has been widely used for the detection of oxidative-reductive compounds in the HPLC analysis.  In the present study, the mechanical condition of the ECD detection is only optimized for three beta-agonists, which is naturally carried out for the HPLC analysis of the target compounds. From this reason, the novelty of the present study is considered to be low so that the paper is not recommended to the publication in Applied Sciences.

Answer: ECD based methods, and in particular the PAD based ones, have been extensively exploited as detection methods in HPLC. In this context, the main point to be considered in the development of such methods is the perfect tune between the mobile phase, also acting as the supporting electrolyte, and the electrode material/electrochemical technique used for the detection of analytes. As a consequence, organic mobile phases used in RP-HPLC are not really compatible with ECD at metal based electrodes. On the contrary, carbonaceous electrodes (including GCEs) possess excellent electrochemical properties in a wide range of working potentials in organic solvents, but these electrodes undergo fouling phenomena, when operate at constant potential, with a time-dependent deterioration of the electrochemical response. To overcome this limitation, we propose for the first time the proof-of–concept of PAD at GCEs (see Analytica Chimica Acta, 894 (2015) 1-6, now ref. 30). The proposed approach allowed the development of a RP-HPLC/PAD method for polyphenols determination in real samples (see Journal of Chromatography A, 1420 (2015) 66–73, now ref. 31). Taking into account that each analyte could have different electrochemical properties at GCE, depending on their electroactive functional groups and the supporting electrolyte (organic mobile phase), a proper electrochemical characterization and optimization of the PAD waveform parameters are required in order to obtain the best detection performance. Therefore, as suggested by the Reviewer, the introduction and conclusions have been modified as follow, to better evidence the improvements of the proposed method with respect to the state of the art.

The paper is also roughly completed with a lot of word hyphen mistakes and the collection of Figures and Tables at the end of the text (3.4 section), which should be improved to distribute to the suitable position between the text sentences.

Answer: Corrections were made accordingly.

Detailed comments are described below:

In the Introduction, it describes that the present method “does not require the derivatization of the analytes or extensive sample pretreatment (page 3, line 3 – 5). However, only three calf urine samples for clenbuterol were measured using the present method as summarized in Table 4, and these obtained values were not evaluated in the results and discussion section. Urine is easy to collect at large volumes so that the reason why the sensitive detection of these three compounds is required should be clearly explained except for the technical points.

Answer: The urine samples analysed were much more than three considering the 20 blank samples for the determination of the selectivity and all the spiked samples used for both the calibration curves in matrix and the recovery studies. In table 4 are summarized the results of samples where was naturally present the clenbuterol, which were discussed in the proper section (see at line 260). Concerning the large volume collection of urine, it is not an easy task during official inspections, and the consequent clean-up steps will require a lot of time, materials, etc., so making the preparation step costly and time-consuming, also considering the huge number of real samples to be processed.

In the Results and discussion section, it describes that “20 independent calf urine samples have been analyzed” (page 5, line 230 – 231). However, the obtained data are not found in this paper.

The recovery test is only applied to clenbuterol at the values of 78.6 – 93.4%, while no data on terbutaline and fenoterol are found in the paper. The experimental result of the recovery test for three target compounds should be clearly summarized in one table, because the simple pretreatment is proposed to the present method.

Answer: Analyses of independent blank samples is aimed at evaluation of selectivity, which is a method performance parameter, in terms of presence of interferences in the elution time window of each analyte. An example of chromatogram of one of the urine blank sample analysed is displayed in figure 4 and compared with a standard mix. Effectively, the sentences are not clear, and then they have been rephrased.

Concerning the recoveries, a new table (now table 4) with recoveries at the tested concentration for each analyte has been added.

In Figures 4 and 5, large amounts of hydrophilic compounds in the calf urine sample are eluted before or near terbutaline and fenoterol peaks, while the pretreatment using the C18 cartridge is completed. The impurity peak interrupts the terbutaline and the fenoterol peaks. This result requires further pretreatment procedures.

Answer: Urine is one of the most complex matrices being plenty of interferences, and in preliminary experiments (data not shown) we tested 4 different conventional sample clean-up methods: SPE-C18, SupelMIPTM SPE, IAC, and Bond Elut Plexa PCX. SPE-C18 and SupelMIPTM SPE gave clean extract in the elution time window of clenbuterol, but only SPE-C18 showed good results for terbutaline and fenoterol, even if an interference was present between the two analyte. In all the other cases, several interferents were present in the time window of the analyte making impossible an accurate quantification.

Table 2 summarizes the validation data including LOD, LOQ, linear range and R value. In the HPLC chromatogram shown in Figures 4 and 5, different peak areas of three target compounds were found in each individual retention time. Increased peak area is detected in the order of fenoterol, terbutaline and clenbuterol, while same concentration of 100 µg/L for each compound is injected into the present HPLC system.  However, in Table 2 the LOD value of terbutetaline is lower than that of fenoterol and clenbuterol, where the LOD value of fenoterol is equal to that of clenbuterol.  Higher LOQ value is also listed in the order of fenoterol, clenbuterol and terbutaline, while each linear range is all same in these compounds.  These disagreements are not understandable for readers.  The range of significant figures should be considered to each value.

Answer: LOD and LOQ depend on the signal but also from the noise that in this case were obtained from the regression parameters, namely slope and standard deviation of residuals of the calibration curve registered in the low concentration range (0.5 – 20 µg/L). Therefore, it is expected that in spite of a high signal response (peak height expressed in mC not area that is expressed in mC·s) a high value of standard deviation of residual produces a high value of LOD or LOQ. However, in the first version of table 2 it was erroneously reported for clenbuterol the LOQ obtained in urine matrix (method limits) instead of that calculated from standard solutions (instrumental limits). Now, method limits have been reported in table 2 in separate columns.

Table 4 summarizes the comparison of analytical values of clenbuterol in calf urine between RP-HPLC/PAD and ELISA. However, the data obtained from ELISA is expressed by “negative” and “positive”, which may difficult to compare the analytical values between these two measurement systems. In the calf urine sample #2, the “negative” in ELISA is corresponded to 0.45 in RP-HPLC/PAD, and in the calf urine sample #3, the “positive” in ELISA is corresponded to 1.5 in RP-HPLC/PAD.  What is the value obtained by RP-HPLC/PAD corresponded to the lowest “positive” value obtained by ELISA?  As to terbutaline and fenoterol, similar experiments should be required.

Answer: The ELISA method used for comparison was a screening method validated for qualitative purposes. Therefore, quantification could not be considered significant and only urine samples with a clenbuterol level above MRL were judged positive. Obviously, by the proposed RP-HPLC/PAD method an accurate quantification is possible for concentration levels equal or above the LOQs.

Other comments

・page 1, line 33: Beta-agonists ---> beta-agonists

・page 1, line 34: Calf urine ---> calf urine

・page 2, line 47: dissi-ness ---> dissiness

・page 2, line 63: gas-chromatography ---> gas chromatography

・page 2, line 67: deter-minations ---> determinations

・page 2, line 68: sim-plicity ---> simplicity

・page 2, line 70: aryletanlaminic ---> arylethanolaminic

・page 2, line 70: fig. 1 ---> Figure 1

・page 2, line 71: am-pero- ---> ampero-

・page 2, line 77: detec-tion ---> detection

・page 2, line 78: prod-ucts ---> products

・page 2, line 83: sur-face ---> surface

・page 2, line 84: continu-ously ---> continuously

・page 2, line 85: amperom-etry ---> amperometry

・page 2, line 86: pro-duce ---> produce

・page 3, line 97: quali-ta- ---> qualita-

・page 3, line 106: acetic acid glacial ---> glacial acetic acid

・page 3, line 132: acetic glacial acid ---> glacial acetic acid

・page 3, line 142: witha ---> with a

・page 3, line 147: min.. (double periods) ---> min.

・page 8, Table 2: β-agonist ---> β-Agonist

Answer: Changes have been made accordingly.

Reviewer 6 Report

The manuscript presents a novel analytical method for the determination of three beta-agonists (clenbuterol, fenoterol, and terbutaline) in urine samples at trace levels by solid phase extraction followed by reversed-phase liquid chromatography and pulsed amperometric detection. The science is sound, the method was validated following European guidelines and the obtained results enabled the quantification of real samples at the trace levels required by current regulations on this topic. Overall, the manuscript is well presented and, in the reviewer's opinion, it is suitable for publication in Applied Sciences journal after addresing various concerns.

Specific comments:

  • Replace "low ppb levels" in the manuscript title for a more appropriate expression, such as "low trace levels" or "low µg/kg levels". 
  • Figure 4. Add the chromatogram of an urine sample spiked with the 3 target analytes. 
  • Figure 5. In the chromatogram of the spiked sample only Clenbuterol is visible, were all the analytes spiked? why the others are not visible in the spiked sample?
  • Table 1. Remove sentence from the template "Tables should be placed in the main text near to the first time they are cited".
  • Table 2. Include instrumental limits (LOQ, LOQ) and method limits (mLOD, mLOQ)

According to these comments and suggestions, the reviewer's overall recommendation is to Accept the manuscript after minor revision.

Author Response

Response to Reviewer 6

The manuscript presents a novel analytical method for the determination of three beta-agonists (clenbuterol, fenoterol, and terbutaline) in urine samples at trace levels by solid phase extraction followed by reversed-phase liquid chromatography and pulsed amperometric detection. The science is sound, the method was validated following European guidelines and the obtained results enabled the quantification of real samples at the trace levels required by current regulations on this topic. Overall, the manuscript is well presented and, in the reviewer's opinion, it is suitable for publication in Applied Sciences journal after addresing various concerns.

Specific comments:

Replace "low ppb levels" in the manuscript title for a more appropriate expression, such as "low trace levels" or "low µg/kg levels".

Answer: The change has been made.

Figure 4. Add the chromatogram of an urine sample spiked with the 3 target analytes.

Figure 5. In the chromatogram of the spiked sample only Clenbuterol is visible, were all the analytes spiked? why the others are not visible in the spiked sample?

Answer: A chromatogram of urine spiked with all the target analytes has been added in figure 5.

Table 1. Remove sentence from the template "Tables should be placed in the main text near to the first time they are cited".

Answer: The sentence has been removed.

Table 2. Include instrumental limits (LOQ, LOQ) and method limits (mLOD, mLOQ)

According to these comments and suggestions, the reviewer's overall recommendation is to Accept the manuscript after minor revision.

Answer: Method limits have been reported in table 2 in separate columns.

Round 2

Reviewer 2 Report

-

Author Response

Text modifications have been carried out accordingly.

Reviewer 4 Report

The manuscript has been substantially improved. In my opinion, it is suitable for publication. 

Author Response

(The authors gave the same response as above.)

Reviewer 5 Report

The present paper is the improved version of the manuscript previously submitted to Applied Sciences.  The paper has been mostly revised according to the reviewer’s comments.  Minor revisions are required as follows:

  1. page 2, line 95 – 101: The sentences are duplicated with those in line 80 – 86.
  2. page 13, line 436, 437: The words are duplicated with those in line 435.

Author Response

(The authors gave the same response as above.)
